# Metformin Improves Ovarian Cancer Sensitivity to Paclitaxel and Platinum-Based Drugs: A Review of In Vitro Findings

**DOI:** 10.3390/ijms232112893

**Published:** 2022-10-25

**Authors:** Giovanni Tossetta

**Affiliations:** 1Department of Experimental and Clinical Medicine, Università Politecnica delle Marche, 60126 Ancona, Italy; g.tossetta@univpm.it; Tel.: +39-0712206270; 2Clinic of Obstetrics and Gynaecology, Department of Clinical Sciences, Università Politecnica delle Marche, Salesi Hospital, Azienda Ospedaliero Universitaria, 60126 Ancona, Italy

**Keywords:** chemotherapy, metformin, paclitaxel, platinum, resistance, ovarian cancer

## Abstract

Ovarian cancer is one of the most dangerous gynecologic cancers worldwide, showing a high fatality rate and recurrence due to diagnosis at an advanced stage of the disease and the occurrence of chemoresistance, which weakens the therapeutic effects of the chemotherapeutic treatments. In fact, although paclitaxel and platinum-based drugs (carboplatin or cisplatin) are widely used alone or in combination to treat ovarian cancer, the occurrence of chemoresistance significantly reduces the effects of these drugs. Metformin is a hypoglycemic agent that is commonly used for the treatment of type 2 diabetes mellitus and non-alcoholic fatty liver disease. However, this drug also shows anti-tumor activity, reducing cancer risk and chemoresistance. This review analyzes the current literature regarding the role of metformin in ovarian cancer and investigates what is currently known about its effects in reducing paclitaxel and platinum resistance to restore sensitivity to these drugs.

## 1. Introduction

Among gynecologic cancers, ovarian cancer is one of the most lethal, showing a high fatality rate and recurrence due to diagnosis at an advanced stage of the disease and the development of chemoresistance, which weakens the therapeutic effects of the chemotherapeutic treatments [1,2,3,4]. In fact, although most patients are initially responsive to the chemotherapeutic treatment with paclitaxel and platinum-based drugs (carboplatin or cisplatin), almost 80% of women relapse due to chemoresistance occurrence with these treatments [1]. Metformin (also known as 1,1-dimethylbiguanide hydrochloride) is a hypoglycemic agent commonly used for the treatment of type 2 diabetes mellitus and non-alcoholic fatty liver disease, but it also shows anti-tumor activity, reducing cancer risk and chemoresistance [5,6]. Importantly, metformin also plays a key role in regulating mitochondrial function since it can inhibit complex I of the respiratory chain [7]. Although paclitaxel and platinum-based drugs are widely used alone or in combination to treat ovarian cancer, chemoresistance occurrence significantly reduces their effects. The chemical structures of paclitaxel, carboplatin, cisplatin, and metformin are shown in Figure 1.

Platinum drugs, such as cisplatin and carboplatin, used alone or in combination with other drugs, are the most used chemotherapeutics against ovarian cancer, and among them, cisplatin (cis-diamminedichloroplatinum II, CDDP) shows the highest efficiency. Cisplatin and carboplatin form both mono-adducts (binding the N7-guanine in DNA), and intra- and/or inter-strand crosslinks [4,8,9]. These DNA structural alterations block DNA synthesis and transcription, causing cell death [4,10]. One of the most important mechanisms of platinum-based drug resistance consists of the binding inhibition between platinum and DNA by activating efflux transporters. In addition, DNA alterations induced by platinum-based drugs can be fixed by activating DNA repair pathways [1]. Another important mechanism of the action of platinum derivatives is associated with the generation of potent cellular reactive oxygen species (ROS) [3,11]. Indeed, several enzymes involved in toxic effect neutralization due to excessive ROS levels have been proposed as therapeutic targets for several malignancy treatments, including those for ovarian cancer [3,12,13,14], since enzyme inhibition is able to improve tumor cell chemosensitivity [15,16,17,18]. Consistently, it deserves to be mentioned that ovarian cancer cells also show increased expression of antioxidant enzymes, which can inactivate the reactive oxygen species (ROS) induced by platinum-based drugs [3,19]. 

Paclitaxel exerts its anticancer effects by binding to the tubulin β-subunit, leading to tubulin polymerization in the absence of guanosine-5′-triphosphate (GTP), which is a factor normally required for microtubule polymerization [20]. Paclitaxel binding to tubulin stabilizes the microtubules, preventing tubulin depolymerization. This process inhibits microtubule shortening during anaphase in the cell cycle by blocking sister chromatid separation, leading to cell death [21]. Several mechanisms of paclitaxel resistance have been found. In particular, paclitaxel, like platinum-based chemotherapies, can be pumped out by efflux transporters, such as the ATP-binding cassette transporter (also known as ABCB1, P-glycoprotein, P-gp, or MDR1) [22,23,24]. Moreover, the antiproliferative effects of paclitaxel can be counteracted by the activation of pro-mitotic factors, such as phosphoinositide 3-kinase/protein kinase B (PI3K/AKT) pathway activation. In addition, paclitaxel can transform B-cell lymphoma 2 (Bcl-2) anti-apoptotic to pro-apoptotic activity by binding to the N-terminal loop of Bcl-2; thus, paclitaxel-resistant cancer cells can counteract this pro-apoptotic activity by increasing the expression of Bcl-2 family anti-apoptotic members [25,26]. It has been found that metformin can induce apoptosis in both primary ovarian cancer cells and SKOV-3 cells by downregulating Bcl-2 and Bcl-xL expression and upregulating Bax and Cytochrome c expression. Moreover, metformin treatment can lead to cell cycle arrest in the G0/G1 and S-phases. Interestingly, the apoptotic effects of metformin can be enhanced by combining metformin with carboplatin and/or paclitaxel, highlighting the important role of metformin in improving the sensitivity to these drugs [27]. 

The aim of this review was to provide an overview of the current literature regarding the role of metformin in ovarian cancer’s response to standard therapy, particularly investigating its effects in reducing paclitaxel and platinum resistance and restoring the sensitivity to these drugs.

## 2. Metformin as Regulator of Cancer Cell Progression and Resistance 

Tunneling nanotubes (TNTs) are small (diameter 50–800 nm) membrane-lined conduits that can form connections between cells and ensure various exchanges, such as the exchange of mitochondria and microRNAs, between cells [28,29,30]. These structures have a potential in vivo role in human malignancies since they can connect tumoral cellular microenvironments with normal ones. Moreover, it has been proven that hypoxic environments can stimulate tunneling nanotube (TNT) formation in both SKOV3 and C200 chemoresistant ovarian cancer cells [31]. Desir and colleagues found that the TNT formation rate of A2780 cells was higher than that of SKOV3 and benign ovarian epithelial (IOSE) cells when co-cultured with chemoresistant C200 and chemoresistant A2780 cells. Moreover, hypoxic conditions increased TNT formation between chemoresistant ovarian cancer cells. Interestingly, metformin treatment significantly decreased TNT formation by suppressing the mTOR signaling pathway, a key modulator of TNT formation [28], under both hypoxic and normoxic conditions in human malignant pleural mesothelioma cells. Interestingly, the authors found that metformin inhibited mTOR through the activation of 5′ AMP-activated protein kinase (AMPK), suggesting that metformin is an indirect inhibitor of mTOR. Importantly, TNT formation was most dramatically inhibited in platinum-resistant SKOV3 and chemosensitive A2780 cells. Thus, TNT formation represents a potential mechanism for intercellular communication in both chemosensitive and chemoresistant ovarian cancer, suggesting that TNTs are a potential therapeutic target in cancer-directed therapy [31].

Autophagy is a self-stabilizing process characterized by a regulated degradation and recycling of cellular organelles and proteins that play a pivotal role in cell protection and cell death since autophagy often precedes apoptosis [32,33,34]. Thus, autophagy is a double-edged sword in tumor development since it can regulate cancer cell resistance to chemotherapeutic agents that promote apoptosis. Interestingly, it was reported that combining metformin with CDDP and methotrexate (MTX) significantly decreased the half-inhibitory concentration (IC50) of CDDP and MTX in the drug-resistant cancer cells SKOV3/CDDP, indicating that cell proliferation was more inhibited when metformin was used with chemotherapeutics than when using chemotherapeutics alone. Importantly, the authors found that metformin increased microtubule-associated protein 1 light chain 3-II (LC3) protein expression, a marker of autophagy, increasing autophagy in SKOV3/CDDP cells compared with that in SKOV3 cells, demonstrating that metformin can sensitize drug-resistant ovarian cancer cells to chemotherapeutic agents by inducing autophagy [35]. Long non-coding RNAs (lncRNAs) are RNA sequences of about 200 nucleotides that play important roles in the regulation of multiple biological activities, including cell proliferation, differentiation, autophagy, and apoptosis. Accumulating evidence shows that the impairment of lncRNA regulation can lead to the development of chemoresistance and metastasis in many types of cancers, including ovarian cancer [3,36,37,38,39]. Different from lncRNAs, microRNAs (miRNAs) are smaller (16–30 nucleotides) noncoding RNAs that can regulate gene expression by targeting specific 3′-untranslated regions (3′-UTR) of target mRNA. Both lncRNAs and miRNAs regulate important cell processes and have been used as markers in many cancerous and non-cancerous diseases [36,37,39,40,41,42,43]. Importantly, these two types of noncoding RNAs are not independent of each other, but there is a cross-talk in the regulation of their expression that allows lncRNAs to regulate miRNA expression and vice versa [40]. Interestingly, it has been reported that metformin inhibits cell viability, migration, invasion, and autophagy and promotes apoptosis in SKOV3 and A2780 paclitaxel-resistant ovarian cancer cells (SKOV3/PR and A2780/PR). In fact, the authors found that metformin significantly reduced lncRNA small nucleolar RNA host gene 7 (SNHG7) expression, but the overexpression of this gene played an important role in inducing paclitaxel resistance in both cell lines. The authors found that metformin treatment could reverse SNHG7-mediated paclitaxel resistance and autophagy in ovarian cancer cells. The authors also proved that in non-small-cell lung cancer (NSCLC), SNHG7 directly bound to miR-3127-5p, an inhibitor of cell proliferation, invasion, and drug resistance [44], and metformin could increase miR-3127-5p levels in paclitaxel-resistant cells by inhibiting SNHG7 expression. Moreover, metformin treatment significantly decreased the tumor growth and autophagy in xenografts of A2780/PR overexpressing SNHG7. Thus, metformin treatment could improve paclitaxel sensitivity by regulating SNHG7/miR-3127-5p-mediated autophagy in ovarian cancer cells [45].

Cancer stem cells (CSCs) are tumor cells characterized by high tumorigenicity, self-renewal capacity, and a high rate of metastases and chemoresistance [46,47]. An interesting study evaluated metformin’s effect during chemoresistance development in A2780 and OAW42 ovarian cancer cells. Interestingly, the authors found that metformin cotreatment significantly reduced cell proliferation and migration and reduced ERK and AKT kinase activation by increasing chemosensitivity to cisplatin and paclitaxel. The authors found that the increased chemosensitivity was due to CSC population reduction in both A2780 and OAW42 cells. In fact, the expression of CD133, a CSC biomarker, and pluripotent genes, such as Oct 4, Sox 2, and Nanog, were found to be significantly downregulated in these chemosensitive cells. The authors found that the levels of two amino acids (taurine and histidine) were upregulated in resistant A2780 cells treated with metformin, demonstrating the metabolic modulation of metformin in reducing CSC cells by increasing taurine and histidine levels. These data are in agreement with previous studies showing an important role of taurine in promoting stem cell differentiation [48,49]. These findings were validated by other authors treating chemoresistant ovarian cancer cells with taurine. In fact, this treatment considerably reduced the cancer stem cell population and chemoresistance in these cells [50].

AXL and TYRO3 receptor tyrosine kinases are two of the three members of the TAM subfamily of receptor tyrosine kinases (RTKs), a family of receptors widely expressed in normal and cancerous tissues that is involved in the modulation of pro-survival and anti-apoptotic signals. These receptors play a key role in tumor cells since they can favor cancer cell survival and proliferation. For these reasons, TAM kinases are potential therapeutic targets in cancer treatment since their inhibition can reduce cancer cell survival, enhance chemosensitivity, and reduce metastasis occurrence [51,52]. An interesting study demonstrated that metformin significantly decreased the viability of sensitive (A2780 and SKOV3) and cisplatin/taxol-resistant ovarian cancer cells (A2780/CDDP and SKOV3/TR cells, respectively) in a dose-dependent manner. Moreover, metformin treatment of ovarian cancer cells significantly decreased both mRNA and protein levels of Axl and Tyro3 in a dose-dependent manner, indicating that metformin suppresses AXL and TYRO3 expression at the transcriptional level. Metformin treatment also reduced proliferation in SKOV3 and taxol-resistant SKOV3/TR cells transfected with AXL and TYRO3 siRNAs, suggesting these two proteins as targets of metformin. Furthermore, metformin significantly reduced the levels of X-linked inhibitor of apoptosis protein (XIAP), an anti-apoptotic molecule, thus favoring apoptosis. Metformin treatment also showed inhibitory effects on Erk and STAT3 phosphorylation in both sensitive and resistant cell lines. Thus, these data show that metformin can sensitize cisplatin/taxol-resistant ovarian cancer cells to these drugs, reducing AXL and TYRO3 RTK expression and favoring apoptosis [53] suggesting a potential use of metformin as anticancer compound alone or in combination with other molecules to target different cell pathways [54,55].

Nuclear factor kappa B (NF-κB) is an important transcription factor involved in the inflammatory and innate immune responses. However, this transcription factor also plays a key role in drug resistance in many cancer types, including ovarian cancer, and it is constitutively active in many cancerous cells [56]. In fact, NF-κB inhibition significantly reduced cell proliferation and induced apoptosis in drug-resistant ovarian cancer cells [57,58]. Interestingly, it has been reported that metformin exhibited antiproliferative effects in the paclitaxel-resistant A2780/PR and cisplatin-resistant A2780/CDDP cell lines. In fact, cisplatin or paclitaxel combined with metformin significantly improved treatment efficiency, reducing the cell proliferation rate in both sensitive and resistant cells. In addition, metformin significantly reduced the NF-κB signaling pathway and cytokine production. Furthermore, metformin further improved the efficiency of these drugs, reducing drug-induced inflammation [59].

Generally, the tumor microenvironment significantly contributes to cancer progression and the chemotherapeutic response [60]. The role of inflammation in the ovarian cancer microenvironment and the effects of metformin in its regulation were also investigated by Xu and colleagues [61]. In this study, the authors found a significant IL-6 overexpression in the stromal fibroblasts of ovarian cancer samples from patients that had been treated with cisplatin. However, the authors found that the ovarian cancer stroma from patients with routine metformin administration exhibited lower IL-6 expression. Interestingly, the authors found that metformin cotreatment significantly reduced IL-6 secretion in the cisplatin-stimulated MRC5 fibroblast cell line and fibroblast-facilitated tumor growth when cocultured with the ovarian cancer cell line SKOV3, as well as in murine xenograft models. Notably, authors found that metformin treatment significantly inhibited IL-6 secretion by suppressing NFκB signaling. This study highlighted a novel mechanism of metformin in suppressing ovarian cancer progression through the modulation of NFκB signaling in stromal cancer fibroblasts, alleviating stromal inflammation in ovarian cancer [61].

Vascular endothelial growth factor (VEGF) is a key regulator of angiogenesis and vasculogenesis that binds to the tyrosine kinase receptors (VEGFRs) on endothelial cells [62]. However, VEGFRs are present not only in endothelial cells but also in neurons, retinal epithelial cells, and tumor cells, suggesting that VEGF also plays a pleiotropic role in non-endothelial cells. In fact, it has been reported that VEGF induces ERK1/2 phosphorylation in cancer cells, promoting tumor progression and migration [63,64,65,66]. Metformin significantly reduced the viability of HO-8910 cells in a time- and concentration-dependent manner. Moreover, metformin inhibited cell viability and induced apoptosis when it was combined with cisplatin. In addition, the expression of phosphorylated (p)ERK1/2, VEGF, VEGFR2, and B-cell lymphoma 2 (Bcl-2) was downregulated when metformin was used as a cotreatment with cisplatin, whereas Bcl-2-associated X (Bax) and caspase-3 expression was significantly upregulated, demonstrating that metformin in combination with cisplatin significantly improved the cisplatin response, inhibiting ERK1/2 activation by VEGF and VEGFR2 downregulation in ovarian cancer cells [67].

Insulin-like growth factor 1 (IGF-1) is a pluripotent growth factor that binds to its receptor IGF1R, inhibiting apoptosis induced by chemotherapy. In addition, it activates the AKT signaling pathway, favoring cancer progression and chemoresistance by inducing multidrug resistance-associated protein 2 (MRP2) expression [68,69,70,71]. MRP2 is an ATP-binding cassette superfamily transporter that is involved in cytotoxic agent effluxes, including anticancer drugs, such as cisplatin [72,73]. An interesting study evaluated the role of metformin in regulating the IGF/IGF1R/AKT/MRP2 axis. In this study, the authors found that the IC50 value of cisplatin in cisplatin-resistant CP70 cells decreased significantly in a concentration-dependent manner. Moreover, metformin significantly increased apoptosis and the percentage of cells in the G0/G1 phase of the cell cycle. Interestingly, metformin significantly reduced the expression of MRP2, IGF1, IGF1R, pIGF1, pIGF1R, AKT, and pAkt proteins. In nude mice, the tumor volumes of cisplatin-treated groups were significantly lower than those of the control group, and this volume was further reduced by cisplatin and metformin cotreatment, indicating a synergic effect of these two substances in inhibiting tumor growth. Thus, metformin can significantly improve the sensitivity of ovarian cancer CP70 cells to cisplatin by inhibiting the IGF/IGF1R/AKT/MRP2 axis [74].

The studies discussed in the previous paragraph and summarized in Table 1 clearly showed that metformin plays a key role in modulating many important processes involved in ovarian cancer progression and resistance.

## 3. Role of Metformin in Regulating the Metabolic Pathway in Resistant Ovarian Cancer Cells

It is known that tumor cells are defective in mitochondrial respiration and mainly dependent on glycolytic metabolism, but this hypothesis is still a matter of debate since this mechanism has not been found in all types of tumors. Although ovarian cancer is one of the most dangerous gynecological cancers worldwide, its metabolic mechanisms are still poorly understood.

An interesting study by Ricci and colleagues evaluated the metabolic asset in cisplatin-resistant ovarian cancer patient-derived xenografts (PDXs) compared with sensitive PDXs and found that an increase in the glycolytic genes, a downregulation of the gluconeogenic axis, and a decrease in pyruvate production were present in resistant PDXs, suggesting an enhanced glycolytic pathway. In addition, they found that the oxygen consumption rate (OCR) and mitochondrial respiration were higher in resistant PDXs than in sensitive PDXs. Interestingly, the authors found that metformin could reverse platinum resistance in PDXs, suggesting an important role of this drug in improving the response to this chemotherapy [75]. This effect of metformin can be explained by the fact that metformin can impair mitochondrial function by suppressing complex I of the electron transport chain [76], reversing the tumor metabolic properties to those of CDDP-sensitive PDX, thus regaining drug sensitivity.

The importance of targeting the metabolic pathways as a therapeutic approach for ovarian cancer and the role of metformin in enhancing the effects of these treatments have also been demonstrated by other authors.

Tumor necrosis factor receptor-associated protein 1 (TRAP1) is a key modulator of cell metabolism and is upregulated in several types of tumors [77]. It has been reported that TRAP1 expression was inversely correlated with both ovarian cancer tumor stage and lower survival. In particular, TRAP1 expression was lower in the more advanced disease grades, and patients with low expression of TRAP1 had lower survival. Moreover, TRAP1 expression was lower in platinum-resistant (PEA2) cells compared with platinum-sensitive (PEA1) cells. Interestingly, TRAP1 silencing in PEA1 cells induced resistance to cisplatin, and resistant cells showed increased oxidative metabolism, indicating increased mitochondrial respiration. Strikingly, the authors found that cisplatin resistance was reversible upon adding metformin to cisplatin treatment, leading to increased cell death. Interestingly, it has also been found that the increased metabolic activity in low TRAP1-expressing cells was accompanied by increased production of inflammatory mediators, such as interleukin IL-6 and IL-8 [78]. Thus, the chemosensitive effect induced by metformin cotreatment was due to its effect of inhibiting mitochondrial function (inhibiting complex I of the respiratory chain).

Hexokinase II (HKII) is a key metabolic enzyme significantly upregulated in many types of cancer, including ovarian cancer [79,80,81]. HKII is strongly correlated with its upstream activator pyruvate dehydrogenase kinase-1 (PDK1), a key regulator of cell proliferation that regulates HKII expression and metabolic activity [82]. Moreover, HKII expression is also regulated by p53, a key tumor suppressor protein frequently mutated in various subtypes of epithelial ovarian cancer and associated with chemoresistance development, tumor progression, metastasis, and adverse clinical outcomes [2,83]. In particular, p53 can downregulate HKII expression by binding to the HKII promoter, leading to decreased glycolysis and increased chemosensitivity in ovarian cancer cells. Interestingly, it has been reported that metformin can competitively inhibit HKI and HKII by mimicking their enzymatic product glucose-6-phosphate [84]. Thus, Han and colleagues investigated the role of metformin on HKII and phosphoPDK1 (pPDK1, the activated form PDK1) expression, along with CDDP-induced apoptosis in the Hey (p53-wt) and OV-90 (p53-mutant) cell lines. They found that metformin treatment significantly enhanced the apoptotic rate in cisplatin-resistant Hey cells. Moreover, metformin treatment significantly decreased HKII and pPDK1 expression in Hey cells but not in p53-mutant OV-90 cells, suggesting that p53 is required for HKII and pPDK1 suppression induced by metformin and CDDP. In addition, metformin significantly decreased the glucose consumption level in Hey cells treated together with CDDP, contributing to increased apoptosis. Thus, metformin treatment can sensitize chemoresistant ovarian cancer cells to cisplatin via the PDK1/HKII pathway, suggesting a potential role of metformin in improving ovarian cancer sensitivity to cisplatin [85].

Xintaropoulou and colleagues found that the expression levels of GLUT1 and HKII were higher in high-grade serous ovarian cancer (HGSOC) than in non-HGSOC. Moreover, GLUT1 expression increased with cancer stage advancement. Interestingly, glycolytic pathway inhibitors, such as STF31 (a pyridyl-anilino-thiazole that alters glycolytic metabolism and binds the GLUT1 transporter [86]) and oxamic acid (a pyruvate analog acting as a competitive inhibitor of LDH [87]), significantly attenuated cell proliferation in platinum-sensitive (PEA1 cell line) and platinum-resistant (PEA2 cell line) HGSOC cell line models in a concentration-dependent manner. Furthermore, the authors showed a synergic effect of STF31 and oxamic acid when combined with metformin, resulting in a significant increase in cell death in both chemosensitive and chemoresistant ovarian cancer cell lines. These findings support the efficiency of targeting the glycolytic pathway in ovarian cancer and further support the role of metformin in enhancing the effects of this type of treatment [88].

The studies discussed in the paragraph above and summarized in Table 2 show that metformin plays an important role in regulating important metabolic mediators in drug-resistant ovarian cancer cells.

## 4. Role of Metformin in Cotreatment with SB203580 or Phenethyl Isothiocyanate (PEITC)

The p38 mitogen-activated protein kinase (MAPK) signaling pathway plays a key role in regulating many cellular processes, including cell proliferation, differentiation, apoptosis, and chemoresistance, in various types of cancers, including ovarian cancer [89,90]. The MAPK family includes many members, but p38, extracellular signal-regulated kinase (ERK), and c-Jun NH2- terminal kinase (JNK) are considered the key players in carcinogenesis and are the most studied [90,91]. Interestingly, Xie and colleagues found that the expression of p-p38 MAPK was significantly increased in cisplatin-resistant SKOV3/CDDP ovarian cancer cells as well as in the primary ovarian cancer tissues. Moreover, treatment with metformin significantly increased the cisplatin sensitivity of SKOV3/CDDP cells. Importantly, the sensitivity to cisplatin was further enhanced when metformin was combined with SB203580 (a p38 MAPK inhibitor) compared with treatment with metformin alone, suggesting an important involvement of the p38 MAPK signaling pathway in cisplatin-resistant onset in ovarian cancer [54].

Natural compounds from plants, fungi, and marine organisms have been widely used worldwide as dietary supplements or natural drugs in traditional medicine. These compounds show numerous beneficial effects, such as antioxidant, anti-inflammatory, and anti-cancer effects [3,14,92,93,94,95,96]. Phenethyl isothiocyanate (PEITC) is an isothiocyanate present in cruciferous vegetables that has shown antioxidant and anticancer activities [97,98]. It has been found that metformin and PEITC significantly inhibit cell growth when used individually in OVCAR3, CAOV3, SKOV3, A2780, and PA-1 ovarian cancer cell lines. Moreover, PEITC induced apoptosis, but metformin mainly showed a growth inhibitory effect. Treatment with either metformin or PEITC significantly increased both total cellular and mitochondrial reactive oxygen species (ROS). Interestingly, metformin and PEITC cotreatment showed a synergistic effect on ovarian cancer cell lines, including the cisplatin-resistant A2780/CDDP line, suggesting that combined metformin and PEITC treatment can also significantly decrease cancer cell growth and induce apoptosis in cisplatin-resistant ovarian cancer cells [55].

## 5. Conclusions and Further Remarks

Ovarian cancer is among the most lethal gynecologic malignancies due to late diagnosis and the occurrence of chemoresistance, which significantly weakens the chemotherapeutic efficiency in these patients [1,2,3,4]. Although metformin is widely used as a hypoglycemic agent to treat patients with type 2 diabetes mellitus and non-alcoholic fatty liver disease, this drug showed important anti-tumor activity in many types of cancer, including ovarian, breast, and endometrial cancer and uterine myomas [99,100,101,102,103,104,105,106,107]. The most accepted hypothesis regarding the mechanism of action of metformin in cancer cells is that this compound is an important inhibitor of respiratory chain complex I, which oxidizes NADH generated through the Krebs cycle [7,108]. Metformin is a low-cost drug and is clinically safe, and its pharmacodynamic profile has been well characterized. Thus, it is an ideal candidate for development as an anticancer compound. However, the epidemiological studies identifying metformin as a potential anticancer drug are difficult to confirm because these studies have included only diabetic patients. Thus, the use of metformin may have significant side effects if used in a population of patients in whom metformin treatment is not required (e.g., non-diabetic patients). However, this compound showed important anticancer effects in various types of cancer, including ovarian cancer. For these reasons, specific clinical trials investigating the effects of metformin in cancer patients should be performed. In particular, it is important to evaluate whether lower concentrations show significant anticancer effects in order to reduce/avoid possible side effects in patients for whom metformin treatment is not necessary. Importantly, this compound significantly restored drug sensitivity in ovarian cancer cells resistant to paclitaxel and platinum-based drugs (see Table 1 and Table 2). This review found that metformin has additional action mechanisms in improving the sensitivity and efficiency of paclitaxel and cisplatin treatment in sensitive and resistant ovarian cancer cells. In particular, metformin could modulate tunneling nanotubes, autophagy, cancer stem cells, and receptor tyrosine kinases. Moreover, metformin showed significant inhibitory effects on the NF-kB, ERK, and AKT signaling pathways. Interestingly, metformin showed a synergic effect in inhibiting cisplatin-sensitive and -resistant ovarian cancer cell growth when combined with SB203580 (a p38 MAPK inhibitor) or PEITC [54,55]. A schematic figure representing metformin action is shown in Figure 2.

In conclusion, metformin has a key function in improving cancer cells’ response to chemotherapeutic agents in both sensitive and resistant ovarian cancer cells. Therefore, specific clinical trials are needed to assess the potential anti-neoplastic effects of metformin and its potential use in the treatment of patients with ovarian cancer in order to improve the outcome of this disease.

## Figures and Tables

**Figure 1 ijms-23-12893-f001:**
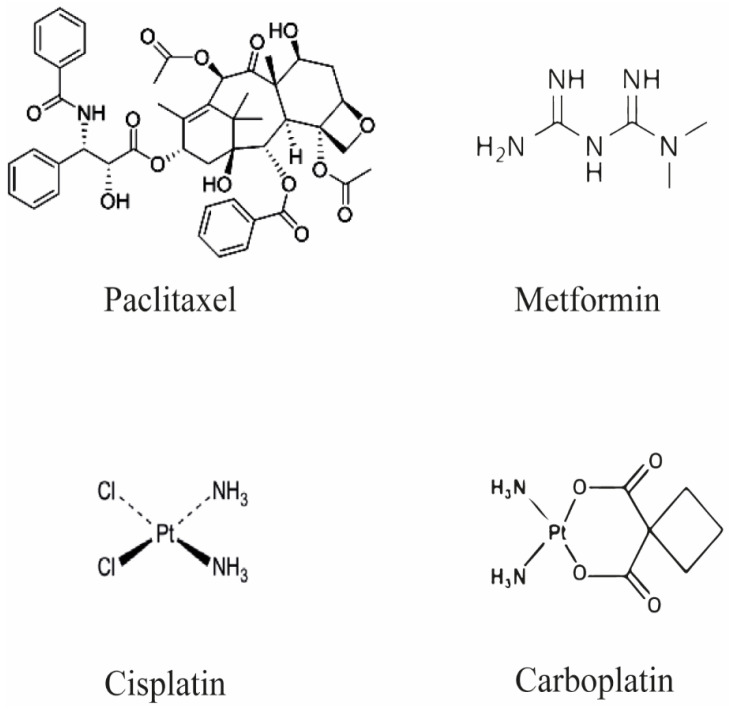
Chemical structures of paclitaxel, carboplatin, cisplatin, and metformin.

**Figure 2 ijms-23-12893-f002:**
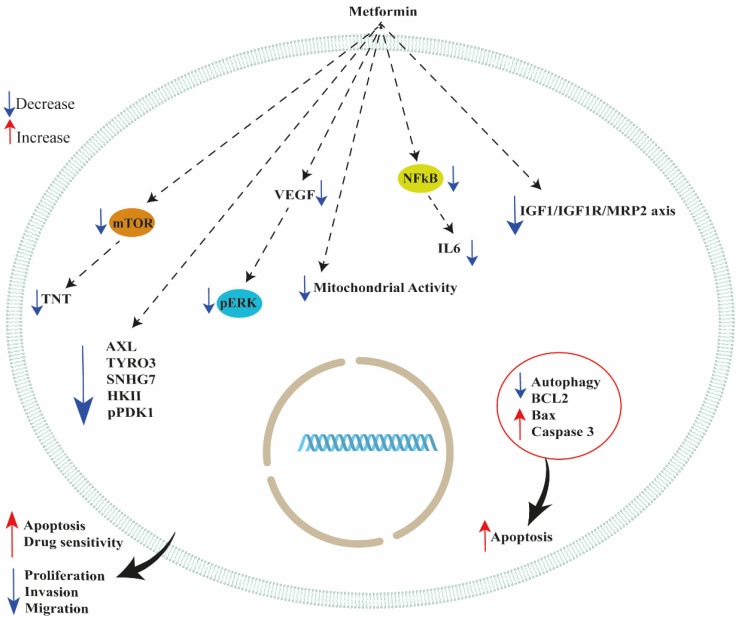
Schematic representation of metformin’s effects on ovarian cancer cells. Tunneling nanotube (TNT); Insulin-like growth factor 1 (IGF-1); IGF-1 receptor (IGF1R); Small nucleolar RNA hostgene 7 (SNHG7); Multidrug resistance-associated protein 2 (MRP2); Extracellular signal-regulated kinase 1/2 (ERK1/2); phospho-AKT (pAKT); nuclear factor kappa-light-chain-enhancer of activated B cells (NF-κB); Vascular endothelial growth factor (VEGF); Hexokinase II (HKII); phosphor- pyruvate dehydrogenase kinase-1 (pPDK1); B-cell lymphoma 2 (Bcl-2); Bcl-2-associated X (Bax); mechanistic target of rapamycin (mTOR).

**Table 1 ijms-23-12893-t001:** Metformin as a regulator of cancer cell progression and resistance.

Model	Target	Results	Ref.
A2780, C200, SKOV3, and IOSE cell lines	Tunneling nanotubes (TNTs)	Metformin decreased TNT formation, suppressing mTOR signaling by AMPK activation.	[31]
SKOV3/CDDP and SKOV3 cells	Autophagy	Metformin increased LC3 expression, inducing autophagy. Combining metformin with DDP or MTX decreased the IC_50_ of CDDP and MTX in the drug-resistant cancer cell lines SKOV3/CDDP.	[35]
Paclitaxel-resistant A2780/PR and SKOV3/PR cells	SNHG7/miR-3127-5p axis	Metformin inhibited cell viability, migration, invasion, and autophagy and promoted apoptosis by reducing SNHG7 expression. Metformin treatment reversed SNHG7-mediated paclitaxel sensitivity and autophagy by increasing miR-3127-5p expression. Metformin decreased tumor growth and autophagy in xenografts of A2780/PR by SNHG7 overexpression.	[45]
A2780 and OAW42 cells resistant and sensitive to cisplatin and paclitaxel	Cancer stem cells (CSCs)	Metformin cotreatment significantly reduced cell proliferation and migration and increased chemosensitivity by reducing the CSC population by increasing taurine levels.	[50]
A2780, SKOV3, cisplatin resistant A2780/CDDP and taxol-resistant SKOV3/TR cells	AXL and TYRO3	Metformin decreased sensitive and cisplatin/taxol-resistant ovarian cancer cell viability. Metformin decreased both AXL and TYRO3 mRNA and protein levels. Metformin treatment reduced ERK and STAT3 phosphorylation in both sensitive and resistant cell lines.	[53]
Paclitaxel-sensitive (A2780) and -resistant (A2780/PR) Cisplatin-sensitive (A2780) and -resistant (A2780/CDDP) cells	NF-κB	The combination of metformin with cisplatin or paclitaxel improved the efficiency of treatment, reducing the cell proliferation rate in both sensitive and resistant cells. Metformin reduced the NF-κB signaling pathway and cytokine production.	[59]
Ovarian cancer samples, MRC5 and SKOV3 cells	NF-κB	Tumor stroma of samples from patients with routine metformin administration exhibited lower IL-6 expression. Metformin cotreatment reduced IL-6 secretion in cisplatin-stimulated MRC5 cells, reducing tumor growth in 3D cocultures with SKOV3 and in murine xenograft models. Metformin inhibited NFκB signaling.	[61]
HO-8910 cells	ERK1/2 activation	Metformin combined with cisplatin inhibited cell viability and induced apoptosis. Metformin reduced pERK1/2, VEGF, VEGFR2, and Bcl-2 expression when used as a cotreatment with cisplatin, whereas the expression of Bax and caspase-3 was upregulated.	[67]
Cisplatin-resistant CP70 cells	IGF/IGF1R/AKT/MRP2 axis	Metformin reduced the IC50 value of cisplatin in a concentration-dependent manner. Metformin increased apoptosis and the cell number in the G0/G1 phase of the cell cycle. Metformin reduced MRP2, IGF1, IGF1R, pIGF1, pIGF1R, AKT, and pAkt expression.	[74]

Insulin-like growth factor 1 (IGF-1); IGF-1 receptor (IGF1R); Small nucleolar RNA hostgene 7 (SNHG7); Multidrug resistance-associated protein 2 (MRP2); Extracellular signal-regulated kinase 1/2 (ERK1/2); phospho-AKT (pAKT); nuclear factor kappa-light-chain-enhancer of activated B cells (NF-κB); cisplatin (CDDP); paclitaxel (PR)

**Table 2 ijms-23-12893-t002:** Studies investigating metformin as a metabolic modulator.

Model	Target	Results	Ref.
Ovarian cancer patient-derived xenografts (PDXs) resistant and sensitive to cisplatin	Mitochondrial activity?	Metformin reversed platinum resistance in cisplatin-resistant PDXs.	[75]
Platinum-sensitive (PEA1) and platinum-resistant (PEA2) cells	Mitochondrial activity	Cisplatin resistance was reversible upon adding metformin to the cisplatin treatment, increasing cell death in TRAP1-silenced cells.	[78]
Hey (p53-wt) and OV-90 (p53-mutant) cell lines	HKII and pPDK1	Metformin enhanced the apoptotic rate in cisplatin-resistant Hey cells and decreased HKII and pPDK1 expression in Hey cells but not in p53-mutant OV-90 cells. Metformin decreased the glucose consumption level in Hey cells treated with CDDP, contributing to increased apoptosis.	[85]
Platinum-sensitive (PEA1) and platinum-resistant (PEA2) cells	Mitochondrial activity?	Metformin synergically increased the effects of STF31 and oxamic acid when used as a cotreatment, resulting in a significant increase in cell death in both chemosensitive and chemoresistant ovarian cancer cell lines.	[88]

Cisplatin (CDDP); Hexokinase II (HKII); phosphor pyruvate dehydrogenase kinase-1 (pPDK1)

## Data Availability

Not applicable.

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
