# Peer review of "Metformin Improves Ovarian Cancer Sensitivity to Paclitaxel and Platinum-Based Drugs: A Review of In Vitro Findings"

_ijms, 2022, doi:10.3390/ijms232112893_

Round 1

Reviewer 1 Report

This is a review article discussing the potential combination use of metformin with chemotherapeutic drugs of platinum and paclitaxel in the treatment of ovarian cancer. I read the paper with great interest. The flow of the paper is good and the authors have explained the potential molecular mechanisms of metformin by summarizing the roles in Tables 1-3. These underlying mechanisms may act synergistically with other combined chemotherapeutic agents. I have some suggestions for the revision of the paper to expand the potential use of metformin on other gynecological tumors/cancers.

1.          Line 33: When mentioning the anti-tumor activity of metformin, the authors may refer to a series of studies that show such anti-tumor activities of metformin on various gynecological tumors/caners including breast cancer [Breast Cancer Res Treat. 2014;145:785-90.], ovarian cancer [Diabetes Metab Res Rev. 2015;31:619-626.], endometrial cancer [Gynecol Oncol. 2015;138:147-153.] and uterine myoma [Ther Adv Endocrinol Metab. 2019 Dec 18;10:2042018819895159.]. These would expand the potential usefulness of metformin in the prevention and/or treatment of other gynecological cancers.

2.          Line 357: Similar to the above suggestion, the authors should discuss to strengthen the potential benefits of metformin on gynecological tumors/cancers.

3.          Platinum-based chemotherapy has also been used to treat brain tumors. It is interesting that metformin use in patients with type 2 diabetes mellitus is also associated with a lower risk of benign [Biomolecules. 2021;11:1405.] and malignant brain tumors [Biomolecules. 2021;11:1226.]. The author may also discuss the potential expansion of the combination use of metformin and platinum-based chemotherapeutic agents for other malignancies.

4.          It would be better if a figure can be shown for the interactive effects between metformin and other chemotherapeutic drugs.

Author Response

Author's Reply to the Review Report:

Line 33: When mentioning the anti-tumor activity of metformin, the authors may refer to a series of studies that show such anti-tumor activities of metformin on various gynecological tumors/caners including breast cancer [Breast Cancer Res Treat. 2014;145:785-90.], ovarian cancer [Diabetes Metab Res Rev. 2015;31:619-626.], endometrial cancer [Gynecol Oncol. 2015;138:147-153.] and uterine myoma [Ther Adv Endocrinol Metab. 2019 Dec 18;10:2042018819895159.]. These would expand the potential usefulness of metformin in the prevention and/or treatment of other gynecological cancers.

I added the studies suggested by the reviewer (references in red).

Line 357: Similar to the above suggestion, the authors should discuss to strengthen the potential benefits of metformin on gynecological tumors/cancers.

I improved the discussion as requested (Lines 359- 372)

Platinum-based chemotherapy has also been used to treat brain tumors. It is interesting that metformin use in patients with type 2 diabetes mellitus is also associated with a lower risk of benign [Biomolecules. 2021;11:1405.] and malignant brain tumors [Biomolecules. 2021;11:1226.]. The author may also discuss the potential expansion of the combination use of metformin and platinum-based chemotherapeutic agents for other malignancies.

I thank the reviewer for the suggestion but this is not the aim of this review. However, I will consider the references suggested by the reviewer for a review that I am currently writhing. 

It would be better if a figure can be shown for the interactive effects between metformin and other chemotherapeutic drugs.

I added a schematic figure to show metformin effect in ovarian cancer cells

Reviewer 2 Report

Dear Giovanni,

I read your review and found it interesting. Metformin is quite debated out there and even though many study it and it's quite a 'hot topic' if I may say, there not enough positive evidence for it's use. Maybe because only a subgroup of patients could benefit from it, or maybe it should be combined with other drugs as well in order to ensure a synthetic lethality and/or prevent the tumor from eventually adapting to it too. What's certain is that we're barely half way there... So keep up the good work, best of luck with your research and I hope you'll find soon some colleagues to help you in your journey.

In the attached pdf you can find my suggestions,

May the Force be with you!

Author Response

Author's Reply to the Review Report  in the PDF file attached 

Round 2

Reviewer 1 Report

Line 352: The author has mentioned “……uterine myoma” but the reference on “metformin’s effect on uterine myoma” has not been appropriately cited.

Author Response

I added the reference as suggested by the reviewer (Reference 107)